# *Pseudomonas aeruginosa* clinical blood isolates display significant phenotypic variability

**Robert J. Scheffler[1,2], Benjamin P. Bratton[1,3,4], Zemer Gitai [1]***

**1** Department of Molecular Biology, Princeton University, Princeton, New Jersey, United States of America, **2** Department of Embryology, Carnegie Institution for Science, Baltimore, Maryland, United States of America, **3** Department of Pathology, Immunology and Microbiology, Vanderbilt University Medical Center, Nashville, Tennessee, United States of America, **4** Vanderbilt Institute for Infection, Immunology, and Inflammation, Nashville, Tennessee, United States of America

* zgitai@princeton.edu

**Data Availability Statement:** The sequencing data has been deposited at GenBank under the BioProject ID PRJNA750497 Database Accession Number.

**Funding:** Funding was provided in part by NIH (DP1AI124669: Z.G., B.P.B., and R.J.S., T32

## Abstract

*Pseudomonas aeruginosa* is a significant threat in healthcare settings where it deploys a wide host of virulence factors to cause disease. Many virulence-related phenotypes such as pyocyanin production, biofilm formation, and twitching motility have been implicated in causing disease in a number of hosts. In this study, we investigate these three virulence factors in a collection of 22 clinical strains isolated from blood stream infections. Despite the fact that all 22 strains caused disease and came from the same body site of different patients, they show significant variability in assays for each of the three specific phenotypes examined. There was no significant correlation between the strength of the three phenotypes across our collection, suggesting that they can be independently modulated. Furthermore, strains deficient in each of the virulence-associated phenotypes examined could be identified. To understand the genetic basis of this variability we sequenced the genomes of the 22 strains. We found that the majority of genes responsible for pyocyanin production, biofilm formation, and twitching motility were highly conserved among the strains despite their phenotypic variability, suggesting that the phenotypic variability is likely due to regulatory changes. Our findings thus demonstrate that no one lab-assayed phenotype of pyocyanin production, biofilm production, and twitching motility is necessary for a *P. aeruginosa* strain to cause blood stream infection and that additional factors may be needed to fully predict what strains will lead to specific human diseases.

## Introduction

In recent years, healthcare-associated infections have become an increasing burden on the American healthcare system [1]. One of the major pathogens that contributes to this problem is the Gram-negative rod-shaped bacteria *Pseudomonas aeruginosa*. While *P. aeruginosa* can be found in the environment, it is also known to infect a wide array of hosts from plants to humans [2–6]. The ability to infect such a diverse group of species is due in part to the large number of virulence factors *P. aeruginosa* utilizes. *P. aeruginosa* virulence regulation uses both

GM007388: R.J.S.). Additional funding provided by the National Science Foundation (NSF PHY-1734030: B.P.B.), and VUMC DSP (B.P.B.). The funders had no role in study design, data collection and analysis, decision to publish, or preparation of the manuscript.

**Competing interests:** The authors have declared that no competing interests exist.

chemical factors such as quorum sensing [7] and mechanical cues such as surface association [8,9] to modulate a large suite of virulence factors. These virulence factors include secreted factors like pyocyanin [10], which is regulated by quorum sensing; motility factors like those required for twitching motility [11], which are regulated by surface association; and structural factors like those involved in biofilm formation that are regulated both mechanically and chemically and notoriously difficult to treat with current antibiotics [12–14]. All these factors make *P. aeruginosa* a significant threat to human health.

In addition to possessing multiple virulence factors, *P. aeruginosa* can cause many different types of infections. For example, in humans *P. aeruginosa* infections can be found in sites including blood, skin, subcutaneous tissue, lungs, urinary tract, eyes, and ears. It is possible that *P. aeruginosa* has so many virulence factors because specific factors are specialized for specific host environments. For example, studies of *P. aeruginosa* strains isolated from the sputum of patients with cystic fibrosis suggested that there are specific pressures applied by this environment that drives mutations of *P. aeruginosa*, causing them to be niche specialists [15–21]. For body sites other than cystic fibrosis lungs, such as the blood, it remains unclear if *P. aeruginosa* acts as a niche specialist or if *P. aeruginosa* virulence factors act combinatorially. If strains act as niche specialists there should be little variability amongst a suite of virulence factors than can be assayed in the lab. Alternatively, clinical and environmental isolates can have genomic similarity while displaying phenotypic heterogeneity [22–24].

Here we addressed genomic and phenotypic heterogeneity of bloodstream infections by investigating a collection of 22 recent clinical strains from bloodstream infections of different patients. We measured pyocyanin, biofilm, and twitching assay to survey virulence traits regulated by different mechanisms. We were surprised to see a wide variability in all three phenotypes tested. To understand the genetic basis of this phenotypic variability, we performed whole genome sequencing on each of the strains. Relative to the substantial phenotypic variability displayed by the strains, the genes required for each behavior displayed significantly less variability. These results suggest that in blood infections, *P. aeruginosa* largely retains the structural genes required for multiple virulence-associated behaviors, that the strength of each of these behaviors is variable, and that no one behavior is necessary or strongly predictive of the ability to cause disease.

## Materials and methods

### Strains and growth conditions

All strains were grown at 37˚C in liquid LB Miller (Difco) on a roller drum at 90 rpm. Solid media growth was done on LB 1.5% agar plates.

The RWJ strain collection contains de-identified clinical isolates from bloodstream infections and were obtained from Dr. Melvin Weinstein at the Rutgers Robert Wood Johnson Medical School.

### Pyocyanin assay

Overnight cultures of the RWJ collection, as well as PA14 and PAO1 as pyocyanin positive controls and the pyocyanin-deficien *phzS* mutant as a negative control, were grown, and cell free supernatant was collected by pelleting cells at 8000 $x\,g$ for 1 min. The supernatant was filtered through a 0.22 μm syringe filter to ensure the absence of cells. Aliquots of 50 μl of each supernatant was transferred to a well of a 96 well plate (Corning 3904) in triplicate. A spectrum ranging from 250–500 nm was taken for each well at 1nm increments on a microplate reader (Tecan). A blank of LB was also analyzed as a negative control and subtracted from the supernatant values. The integrated peak area from pyocyanin absorbance (306–326 nm) was

calculated to determine the amount of redox active phenazines, primarily pyocyanin, which absorb light in this region.

## Biofilm assay

The static biofilm assay protocol from [25] and [26] was used. Briefly, overnight cultures of the RWJ collection, as well as PA14 and PAO1 as biofilm positive controls, were diluted 1:100 into M63 minimal media supplemented with 1 mM magnesium sulfate, 0.2% glucose, and 0.5% casamino acids. Aliquots of 100 μl of each strain was placed into a round bottom 96 well plate (Corning 3788) in triplicate and grown overnight at 37˚C. After overnight incubation, cultures were poured off and the plate was washed by submerging the plate into water. Each well was then stained with 125 μl of 0.1% (w/v) crystal violet solution in water. The plate was incubated for 10 minutes at room temperature after which the stain was poured off and the plate was rinsed by submersion in water. The remaining stain was solubilized with 125 μl of 30% acetic acid in water and incubated for 10 minutes at room temperature. The solubilized crystal violet was transferred to a new flat bottom clean 96 well plate (Corning 3370) and the absorbance of the wells was taken at 550 nm on a microplate reader (Tecan) for quantification.

## Twitching assay

Individual colonies of cells of the RWJ collection, as well as PA14 and PAO1 as twitch positive controls, were picked with a 10 μl pipette tip and stabbed through the agar of a LB 1.5% agar plate and placed at 30˚C for 4 days. After 4 days, the agar was gently removed from the dish and a sufficient volume of 0.5% (w/v) crystal violet in water was added to the plate until the surface covered. After 5 minutes of staining, the crystal violet solution was removed, and the plate was washed 3 times with water. The resulting crystal violet stained twitch rings were imaged using a Canon EOS Rebel T1i (Lake Success, NY) and measured in FIJI [27–29].

## Statistical analysis

A unequal variance two-tailed Student's T-test with the Bonferroni correction was used to determine statistical significance ($p_{adj} < 0.05$ *, $p_{adj} < 0.01$ **) using MATLAB (R2020a, Math-Works, Natwick MA) (Fig 1).

The pairwise correlation coefficients were computed using MATLAB (R2020a, Math-Works, Natwick MA). When multiple hypothesis tests were performed (Fig 2) the pValue was corrected with the Bonferroni correction for multiple hypothesis tests. None of these comparisons failed to reject the null hypothesis (no correlation) in favor an alternative hypothesis (any non-zero correlation).

## Whole genome sequencing and assembly

Overnight cultures of the RWJ collection, PA14, and PAO1 were grown and the genomic DNA of each strain was collected using the Qiagen DNeasy Blood and Tissue kit (cat no 6950). Genomic DNA was taken by the Princeton Genomics Core Facility and processed for next generation sequencing fragmenting and barcoding the samples for sequencing. A single lane of the MiSeq (600 nt) was used to sequence the strain collection.

After demultiplexing, raw reads were checked for quality control and barcode removal using the Trim Galore! 0.6.3. The resulting reads were then assembled into scaffolds using Unicycler 0.4.8.0 [30]. ORF calling in these scaffolded assemblies was performed by Prodigal 2.6.3 [31] and these protein and nucleotide databases were searched for relevant matches to a hand curated list of proteins of interest using blast+ [version 2.7.1]. Using blastp, we found the

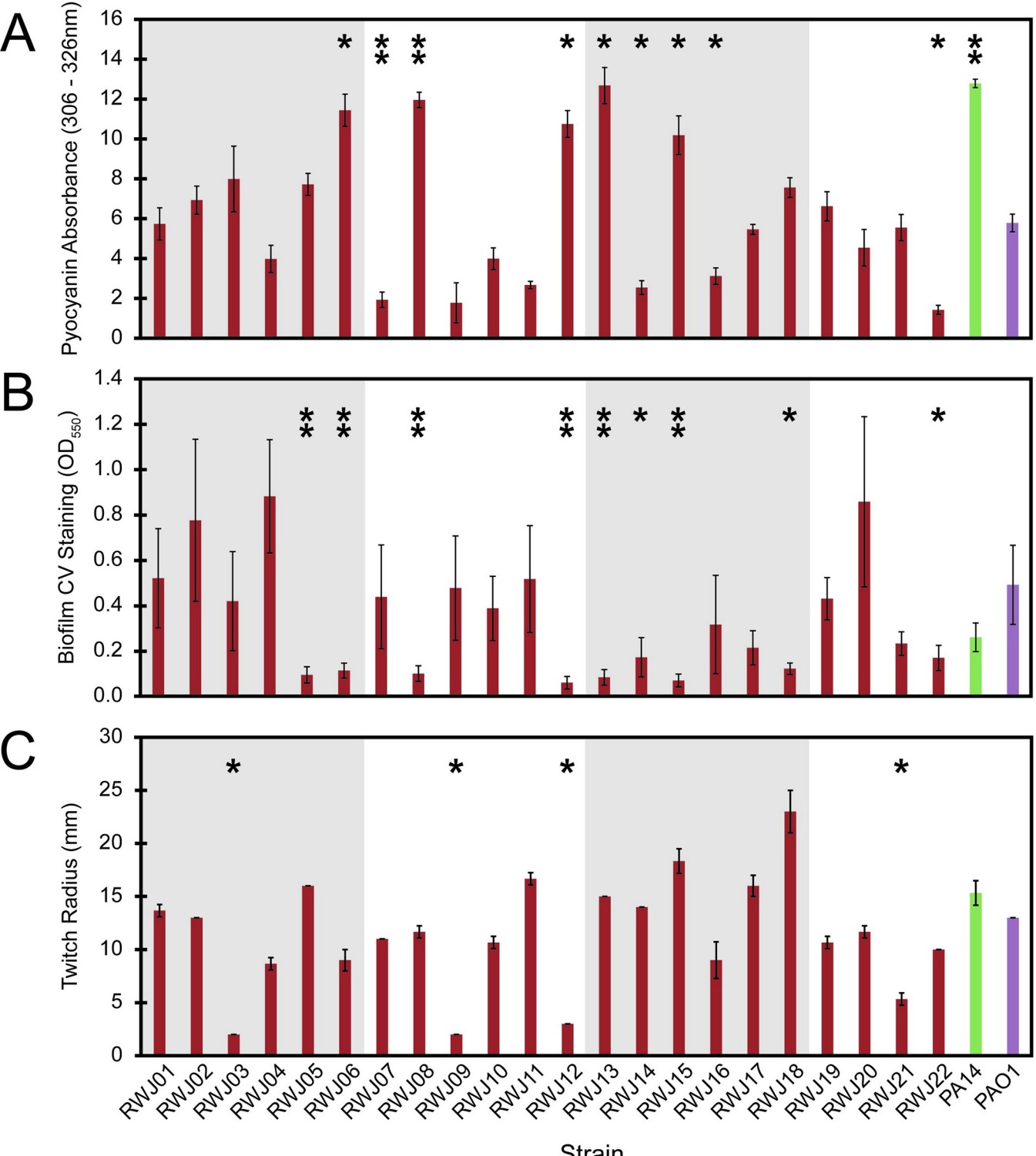

**Fig 1. Virulence phenotype assays vary widely across 22 *Pseudomonas aeruginosa* bloodstream strains.** (A) Pyocyanin absorbance integrated from 306–326 nm. Mean and standard deviation shown from 3 biological replicates (B) Crystal violet staining of total biofilm production as determined by absorbance at 550 nm. Mean and standard deviation shown from 3 biological replicates. (C) Twitching radius determined after crystal violet staining, measured in mm. Mean and standard deviation shown from 3 biological replicates. A Student's T-test with the Bonferroni correction was used to determine statistical significance ($p_{adj}$ <0.05 *, $p_{adj}$ <0.01 **). Strains were obtained from Robert Wood Johnson medical school and are listed as RWJ01-RWJ22 and colored in red. Reference lab strains show are PA14 (green) and PAO1 (purple).

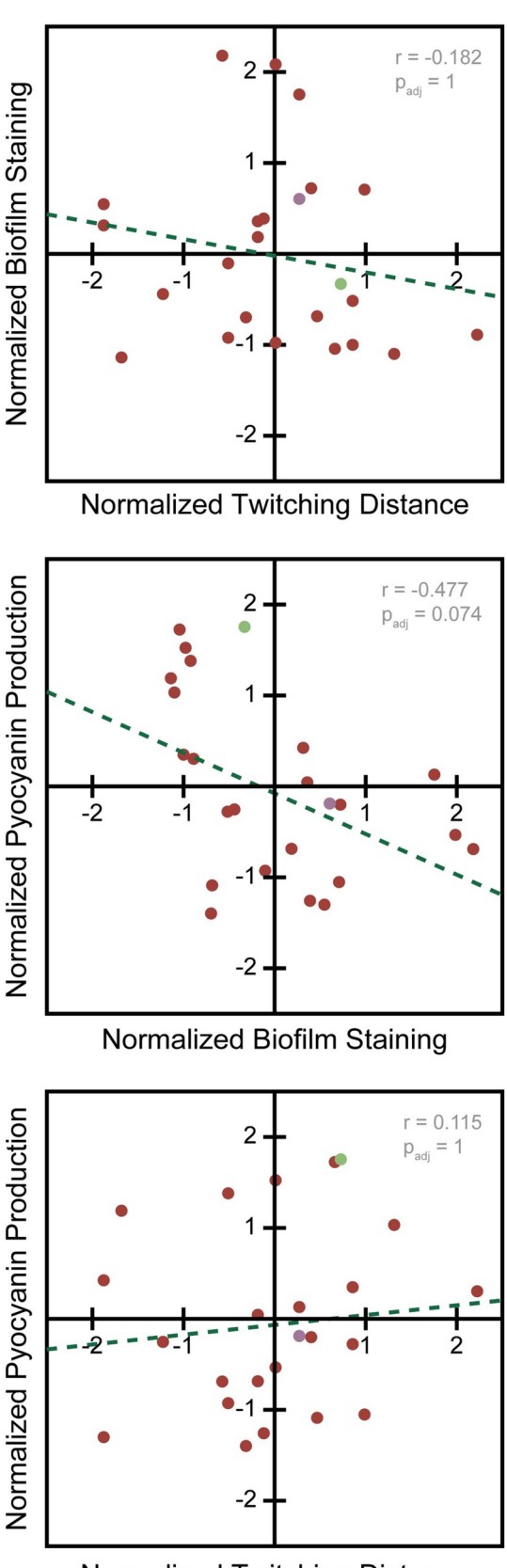

**Fig 2. 2D plots of normalized virulence data show no significant correlation.** (Top) Comparison of twitching distance and biofilm staining. Means of biological triplicates shown. Correlation $p_{adj}$ = 1. (Middle) Comparison of biofilm staining and Pyocyanin absorbance. Means of biological triplicates shown. Correlation $p_{adj}$ = 0.074 (Bottom) Comparison of twitching distance and pyocyanin production. Means of biological triplicates shown. Correlation $p_{adj}$ = 1. RWJ strain data shown in red, PA14 data shown in green, PAO1 data shown in purple.

ORFs with the highest alignment score to the protein of interest (S2 Table). We then concatenated the nucleotide sequence for those entire ORFs to generate a super-gene. In order to generate the phylogenetic trees, multiple sequence alignment was performed by the combination of mafft [32], BMGE [33], and phy-ml [34]. As needed, MATLAB R2019a was used as general purpose scripting language to manage the semiautomated components of the pipeline.

For determining the similarity of proteins as seen in Fig 4 and S1 Fig we queried the assembled scaffold ORF databases for the virulence associated genes (S2 Table) using the published PAO1 reference genome for the best alignment protein. We then took the full ORF result of that search and used it to query back onto the PAO1 reference genome. If the reverse blast did not produce the starting gene query, we eliminated that gene from that strain in our table as we considered this to be an off target match. Similarity scores were calculated either as the alignment score divided by the perfect alignment of the template (S1 Fig) or of the resulting hit (Fig 4) using a BLOSUM62 scoring matrix. This method reduces the penalty for ORFs that run into the end of a contig and are therefore artificially short due to an assembly error instead of a biological difference.

## Results

### Clinical strains show variability in multiple virulence-associated phenotypes

We obtained a collection of *P. aeruginosa* strains from bloodstream infections from the Robert Wood Johnson Hospital to investigate variability amongst strains from the same infection site. This collection contained 22 strains that had been previously isolated from the blood of 22 different patients. When the strains were plated on standard lab LB 1.5% agar plates there was a large amount of variability in colony morphology. 7 of the strains formed rugose colony biofilms while the other 15 formed more mucoid colony biofilms. In addition, 2 of the strains produced blue-green coloration of the agar near the plated bacteria indicating a large quantity of pyocyanin. When the strains were grown in standard LB liquid overnight, 5 of the cultures were a deep blue green, indicative of high pyocyanin production, while another 5 were a pale blue green color. The variability of both the colony morphology and the liquid growth culture colors led us to more rigorously quantify phenotypic variability amongst the strains. As positive control strains for each assay tested, we also measured the phenotypic response of PA14 and PAO1 as they have been well characterized by the field.

To measure pyocyanin production in a lab environment, we collected cell free supernatant and analyzed it by UV-Vis spectroscopy, measuring absorbance from 250–500 nm. The spectrum from a blank well containing only LB was subtracted from each of the sample spectra and the integrated peak area from pyocyanin absorbance (306–326 nm) was calculated. Confirming our initial qualitative observations, there were a wide range of pyocyanin levels ranging from ~2–13 integrated absorbance units (Fig 1A). PA14 showed pyocyanin levels similar to the highest pyocyanin producers of our strains, while PAO1 showed pyocyanin levels in the middle of the range produced by our collection. These results demonstrated that despite the fact that pyocyanin production has been previously correlated with virulence and pathogenicity in other infection sites (4, 10), *P. aeruginosa* can mount a successful blood infection even

when the pyocyanin level, in a mono-cultured test tube setting, nears the bottom of our detection limit.

Another phenotype we sought to further quantify in our collection was biofilm formation. In order to measure the level of biofilm quantitatively we utilized a crystal-violet based microtiter plate biofilm formation assay [25,26]. This assay colorimetrically measures the amount of crystal-violet bound by the biofilm matrix. The amount of biofilm produced ranged from 0.06 to 0.88 (Fig 1B), with some strains making more biofilm than both PAO1 and PA14 and some making less than both PAO1 and PA14. Thus, biofilm production was also highly variable in the collection.

The final phenotype we investigated was the ability of *P. aeruginosa* to twitch, another behavior that has also been implicated as important for pathogenesis. Twitching is a surface associated motility phenotype where cells are able to move along the surface underneath soft agar. This phenotype is visualized by crystal violet staining of the resulting biofilm spread to show the furthest extent of motility. After 4 days of twitching at 30˚C, we saw that while most of the strains twitched to a similar extent as PA14 and PAO1, there were 3 strains that appeared to be completely deficient in twitching and 4 strains that twitched further than PA14 (Fig 1C). Together with our pyocyanin and biofilm assay results, these findings reinforced the conclusion that individual strains from bloodstream infections are highly variable with respect to all lab phenotypes assayed despite the fact that all three phenotypes have been implicated as important for pathogenesis.

## Virulence phenotypes show no significant covariance

Since each of the virulence phenotypes assayed showed significant variability, we sought to determine if any of them covary, which would indicate that they are coregulated, or if they vary independently, which would suggest that they might function combinatorially. Many virulence-related phenotypes can be co-regulated. For example, quorum sensing regulates both pyocyanin production and biofilm formation [35–37], while surface sensing regulated both biofilm formation and twitching motility [8,9,11,38]. However, in our analysis, pairwise comparisons of each of the phenotypes across the strains revealed no significant correlations (Fig 2, $p_{adj} > 0.05$). Twitching motility showed almost no correlation whatsoever with either pyocyanin production or biofilm formation (Fig 2). Meanwhile, pyocyanin production and biofilm formation displayed a slightly negative correlation, but this was not statistically significant (Fig 2 Middle, $p_{adj} > 0.05$). For this analysis, all data were normalized so that the phenotypes were on the same scale for each comparison (Fig 2).

While there was no significant covariance amongst the measured phenotypes, we also observed that there was no strain that was low across all the assays. Strains such as RWJ09 and RWJ12 were low in two of the assays, but high in a third (Fig 1). Other strains like RWJ01 and RWJ02 had no assays in which they were low (Fig 1). All of these data suggest that there is no "stereotypical" phenotype amongst the collection, but that each of the strains examined retained at least one of the pathogen-associated phenotypes tested.

## Phylogenetic clustering and phenotypic clustering differ

The phenotypic variability of the collection in the biofilm, pyocyanin, and twitching assays, led us to investigate the genomes of all the strains by whole genome sequencing. In brief, we used Illumina 2x250 paired end nextgen sequencing. From these reads we assembled scaffolds using Unicycler 0.4.8.0 [30] and called ORFs using Prodigal 2.6.3 [31]. For details on assembly statistics see S1 Table. As a control for the efficacy of our bioinformatic pipeline, we also

resequenced PA14 and PAO1 and used the same assembly pipeline to assemble the genomes *de novo*.

Following assembly of each strain's genome, we used the housekeeping genes as a basis for phylogenetic analysis with the published PAO1 reference genome as a search template. Specifically, we generated a phylogenetic tree based on the distances for the combination of nine housekeeping genes (*dnaE*, *guaA*, *gyrB*, *mreB*, *mutS*, *ppsA*, *recA*, *rpoB*, and *rpoD*). Interestingly, we found several distinct phylogenetic clusters within our strains: RWJ05, RWJ15, RWJ11, and RWJ03 clustered with one another and with PAO1; RWJ02, RWJ10, RWJ14, RWJ08, and RWJ12 formed a distinct cluster also similar to PAO1; RWJ04 formed a distinct cluster with PA14; RWJ06, RWJ22, and RWJ18 form an additional distinct cluster; and RWJ13, RWJ21, RWJ16, RWJ19, RWJ09, RWJ17, RWJ20, RWJ07, and RWJ01 were dissimilar from both one another and from PAO1 and PA14 (Fig 3A).

We noted that strains with similar genomes often exhibited different virulence-associated phenotypes. For example, RWJ04 phylogenetically clustered with PA14 but showed much lower twitching and pyocyanin production and higher biofilm formation than PA14 (Figs 1 and 3A). To examine the relationship between genomic and phenotypic similarity more systematically, we normalized the data from our phenotypic assays and used it to cluster the strains based on their phenotypes (Fig 3B). We observed that the clustering of the phenotypic data looked different from the phylogenetic clustering (Fig 3A and 3B).

To further compare the phenotypic and phylogenetic clusters we used the phylogenetic clustering order to reorder the phenotypic data. These reordered data were then put into a heat map (Fig 3C). The lower left portion of the heatmap is colored by the phylogenetic distance between two strains and the upper right is colored by the phenotypic distance. The lack of symmetry in this resulting heat map shows the lack of correlation between measuring the phenotypic and phylogenetic distances (Fig 3C). Since clustering adds an extra layer of factors into the data, we also plotted the distances between strains from both sets of data against each other and determined the correlation of these distances (Fig 3D). The plotted distances confirmed that there is no significant correlation between phylogenetic and phenotypic distances.

## Most virulence phenotype-associated genes are highly conserved across the collection of strains

Our phylogenetic analysis was based on similarity across a set of housekeeping genes, but the virulence-associated phenotypes we assayed are known to be driven by specific sets of genes. We thus investigated whether there were specific genes that could explain some of the phenotypic variability in pyocyanin production, biofilm formation, and twitching motility. We first defined a distinct set of genes that have previously been associated with each of our three phenotypes. Using a custom MATLAB script, we determined the protein sequences of each of these genes of interest in each of the strains, as well as our assemblies of the reference strains PA14 and PAO1. This analysis included a total of 129 genes implicated in at least one of the three virulence-associated phenotypes. The search templates for these genes came from the published PAO1 reference genome [39]. Each of these homolog ORFs was given a similarity score compared to the reference protein (Figs 4, S1 and S2). Due to the read length and depth of coverage of our reads not every gene could be identified in every strain. If a gene was not identified in any strain, that gene was removed from further analysis. We also removed any ORFs that matched something other than the original template when realigned back against the reference PAO1 genome. Additionally, for ORFs that ran into the edge of the scaffolds, we computed similarity scores for the portion of the protein that was represented in our assembly.

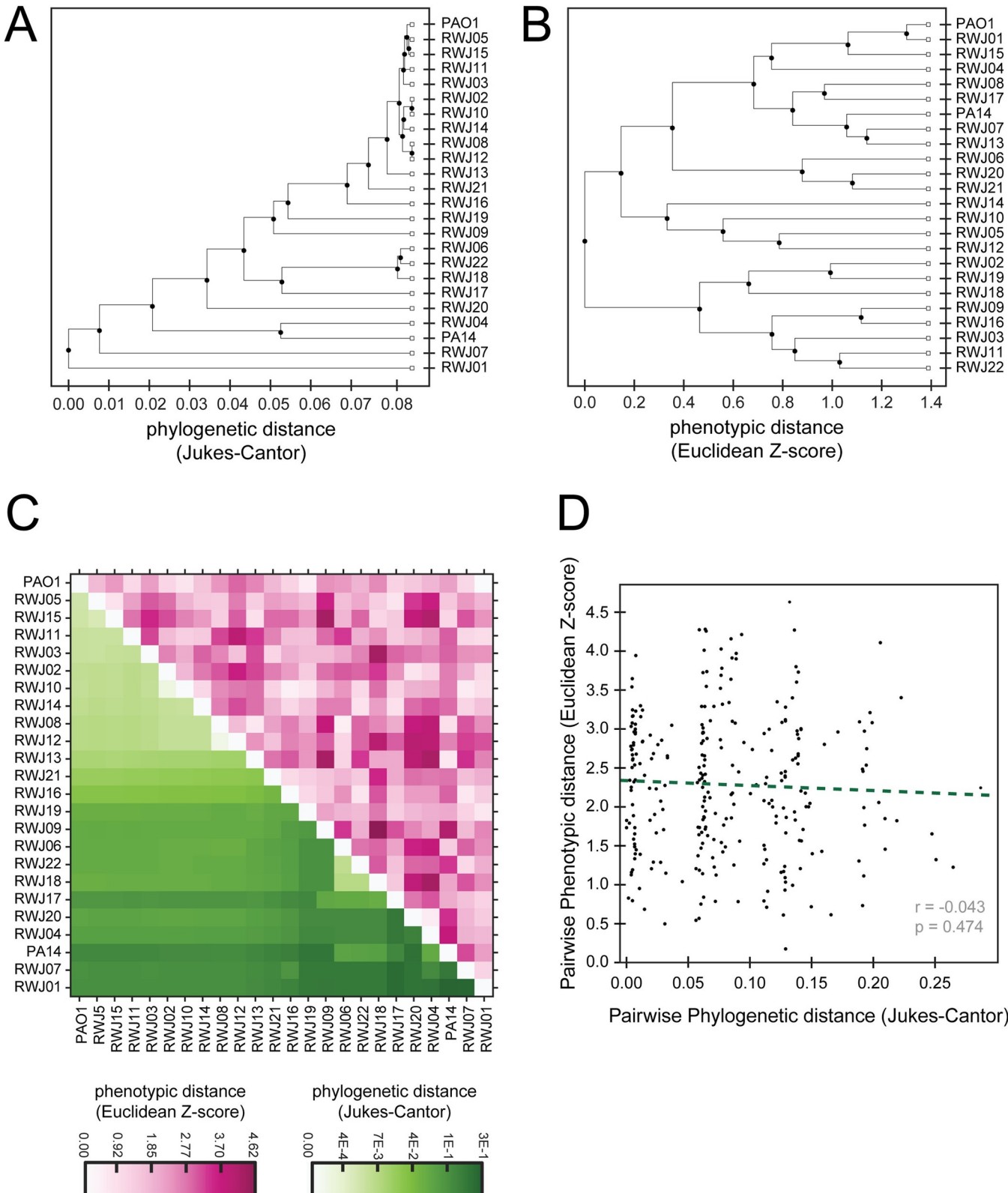

**Fig 3. Clustering of phylogenetic data and phenotypic data are not correlated.** (A) Phylogenetic tree of the RWJ samples, PA14, and PAO1 based on the genes *dnaE*, *guaA*, *gyrB*, *mreB*, *mutS*, *ppsA*, *recA*, *rpoB*, and *rpoD*. (B) Phenotypic clustering based on the 3 virulence assays tested. (C) The clustering order from the phylogenetic tree used to reorder the phenotypic data. White to dark green color used to display phylogenetic distance. White to dark pink color used

to display phenotypic distance. (D) Pairwise phylogenetic distance compared with the pairwise phenotypic distances. No statistically significant correlation observed.

We first examined the biosynthetic genes involved in pyocyanin production. We found that most of the pyocyanin genes searched for shared high homology amongst all the strains (Fig 4A). For example, a low pyocyanin production strain like RWJ09 exhibited similar homology

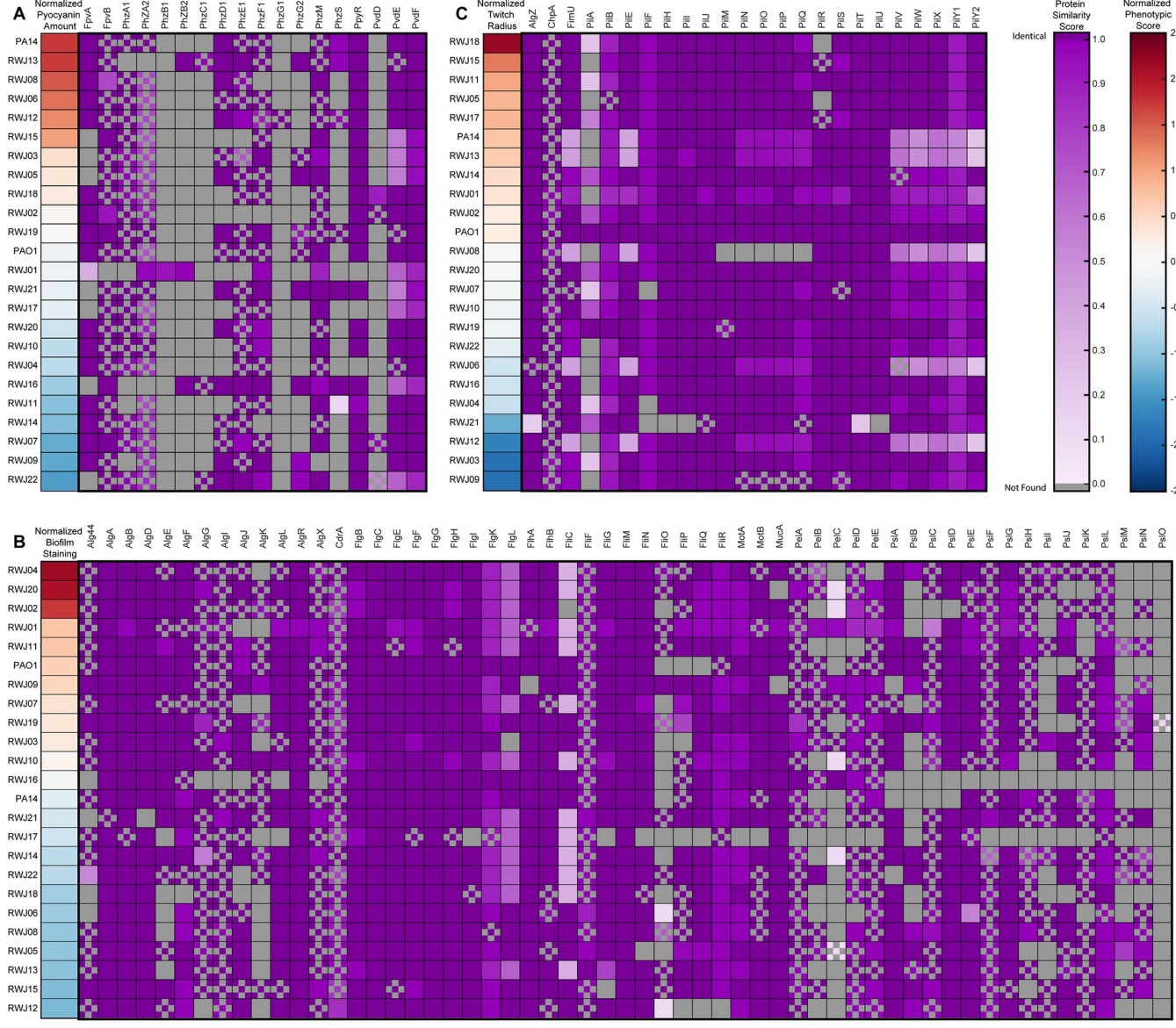

**Fig 4. Protein similarity amongst strains shows high homology amongst most genes.** Genes involved in the (A) pyocyanin and pyoverdine, (B) biofilm, (C) twitching were compared for similarity. (Left) Strains are ordered by their normalized phenotypic score from dark red indicating high to dark blue indicating low. (Right) Similarity scores between the gene in published PAO1 reference genome and ORF from our assembly. Dark purple on indicates 100 percent similarity with decreasing saturation indicating decreasing similarity. Gray boxes indicate that the gene was not found in the scaffolds of the strain and could indicate a loss of the gene or be the result of incomplete assembly of contigs. Checkerboard pattern indicates that the ORF was found running up to the edge of a contig and was scored based on the portion of the protein present. PA14 and PAO1 comparisons shown are based on our de novo assembly of these strains to provide a baseline for our sequencing and assembly pipeline. For scores calculated based on full length template see S1 Fig.

in pyocyanin production genes to RWJ13, which had high pyocyanin production. Similarly, RWJ06, RWJ19, and RWJ07 each had nearly perfectly conserved pyocyanin genes yet exhibited high, moderate, and low pyocyanin production, respectively (Fig 4A). Thus, analysis of pyocyanin-related genes failed to identify a clear genetic signature associated with pyocyanin production (or lack thereof).

Analysis of biofilm production genes produced similar results to those of pyocyanin production, with high overall conservation of genes required for each phenotype and no clear genetic predictors of phenotypic strength (Fig 4B and 4C). For example, RWJ16 produced moderate biofilm while potentially lacking many biofilm genes whereas RWJ15 has high similarity to almost all biofilm associated genes but limited biofilm formation (Fig 4B). Meanwhile, FliC, the structural flagellin component, showed low homology in roughly half the strains, but this was not correlated to any particular level of biofilm formation (Fig 4B).

Similarly, when looking at genes involved in twitching, the variability of the few genes that displayed significant variability in their homology, like *pilA* and *pilB*, did not correlate with twitching motility distance (Fig 4C). We were able to identify strains that were lacking *pilA* or had low *pilA* homology but high twitching (RWJ18, RWJ15, RWJ11). Furthermore, RWJ03 and RWJ07 had similar homology profiles but differed in their twitch radii (Figs 1C and 4C).

The lack of correlation between the presence of genes required for a given phenotype and the strength of the phenotype could be the result of changes in regulatory proteins that modulate the levels of the genes responsible for the differences in phenotypes. We thus decided to examine quorum sensing genes, as these pathways are known to regulate genes involved in multiple virulence-associated pathways [7,35,36]. However, analysis of the quorum sensing gene sequences showed the same pattern as the other genes examined, with high overall homology and no clear correlation to any specific phenotype (S2 Fig).

## Discussion

Due to *P. aeruginosa*'s ability to colonize and infect a wide array of hosts and body sites, we were interested if strains isolated from the same body site, in particular the bloodstream, of different patients would show variability in their phenotypes. There have been many studies on *P. aeruginosa* virulence factors implicating pyocyanin production, biofilm production, and twitching motility as important mediators of pathogenesis [4–6,40–45]. We observed high variability amongst the phenotypes we assayed, which is consistent with other studied that found high variability amongst isolates from other clinical sites such as urinary tracts and cystic fibrosis lungs [22–24]. It is possible that none of the three specific phenotypes we assayed are important for blood infections or are only induced in blood-like environments. Alternatively, we suggest that perhaps these virulence-promoting traits are partially redundant and can function synergistically when insufficient on their own. While one could continue to measure additional virulence factors, our survey of pyocyanin, biofilm, and twitching account for both mechanically- and chemically-regulated virulence factors. We note that we identified strains with low levels of each of the three individual phenotypes, but no strains in which all three phenotypes were low. Even strains such as RWJ16, RWJ21, and RWJ22, which were below average in all three normalized assay scores, had at least one phenotype at a moderate level. As such, our data support the combinatorial model as there is no one virulence-associated phenotype necessary to cause disease in the blood but all strains exhibited at least moderate levels of at least one phenotype tested.

To address possible genetic sources of the phenotypic variability observed, we performed whole genome sequencing of our bloodstream infection strains. We first established that the phenotypic and phylogenetic similarities across the strains were uncorrelated, as strains that

were phylogenetically similar were phenotypically dissimilar. These results suggest that these phenotypes are not a product of any singular *P. aeruginosa* subtype amongst our collection from a particular body site.

Phenotypic variability can generally be attributed to structural or regulatory changes in the genome. We thus analyzed the conservation of >125 genes previously implicated in structural or regulatory functions associated with each of the three phenotypes investigated. Similar to other bloodstream isolate surveys [46,47], we found that most of these genes were highly conserved among our collection. In addition, the variability amongst genes that were not highly conserved did not correlate with the phenotypic variability, suggesting that these genetic changes could not explain the phenotypic changes. These data suggest that there may be additional factors regulating these behaviors such as the microenvironment of the bloodstream [46] or that their regulation is mediated by non-structural genetic changes, such as those associated with promoters, translation, or mRNA stability [48].

A possible confounding factor in interpreting the phenotypic variability we assayed is that growth conditions in the lab differ from those in the host. It is possible that the when the bacteria are in a bloodstream environment, virulence factor expression is modulated in a manner we did not detect in our assays. Future work to better mimic the complex environments bacteria encounter in a host will be needed to address this possibility [49–53]. As we do not have information regarding the patient's outcome nor the specifics about their disease state in our deidentified samples we cannot assess how the phenotypic variability we saw in the lab assays compares to disease severity. There has been some success in identifying genomic signatures that are predictive of disease in mice [54–56].

Our work underscores the importance of multilevel investigations to determine the level of variability in strains and suggests that there might not be a "stereotypical" strain that defines a specific type of infection. While anti-virulence strategies have been proposed as a way to fight the rise in antibiotic resistance [57,58], these data suggest that in order to fight specific virulence factors we first need to know which ones are displayed by different strains at different infection sites. These findings underscore the need for quick diagnostic virulence assays to enable the success of anti-virulence strategies.

## Supporting information

**S1 Fig. Protein similarity using full length template compared to ORFs.** Genes involved in the (A) pyocyanin and pyoverdine, (B) biofilm, (C) twitching, and (D) quorum sensing were compared for similarity. (Left) Strains are ordered by their normalized phenotypic score from dark red indicating high to dark blue indicating low. (Right) Similarity scores between the gene in published PAO1 reference genome and ORF from our assembly. Dark purple on indicates 100 percent similarity with decreasing saturation indicating decreasing similarity. Grey boxes indicate that the gene was not found in the scaffolds of the strain and could indicate a loss of the gene or be the result of incomplete assembly of contigs. Checkerboard pattern indicates that the ORF was found running up to the edge of a contig.
(PDF)

**S2 Fig. Protein similarity of quorum sensing genes.** Genes involved in the quorum sensing were compared for similarity. Dark purple on indicates 100 percent similarity with decreasing saturation indicating decreasing similarity. Grey boxes indicate that the gene was not found in the scaffolds of the strain and could indicate a loss of the gene or be the result of incomplete assembly of contigs. Checkerboard pattern indicates that the ORF was found running up to the edge of a contig and was scored based on the portion of the protein present.
(PDF)

**S1 Table. Assembly statistics.**
(PDF)

**S2 Table. Genes of interest, their PAO1 reference name, and their associated biological activity category.**
(PDF)

## Acknowledgments

We thank the Genomics Core Facility of the Lewis-Sigler Institute for Integrative Genomics performing the sequencing, the Gitai lab for their support, and Dr. Karina Gutierrez Garcia with advice on assembling the scaffolds.

## Database accession number

The sequencing data has been deposited at GenBank under the BioProject ID PRJNA750497 Database Accession Number.

## Author Contributions

**Conceptualization:** Robert J. Scheffler, Benjamin P. Bratton, Zemer Gitai.

**Formal analysis:** Robert J. Scheffler, Benjamin P. Bratton.

**Funding acquisition:** Zemer Gitai.

**Investigation:** Robert J. Scheffler, Benjamin P. Bratton.

**Methodology:** Robert J. Scheffler, Benjamin P. Bratton.

**Software:** Robert J. Scheffler, Benjamin P. Bratton.

**Supervision:** Zemer Gitai.

**Visualization:** Robert J. Scheffler, Benjamin P. Bratton.

**Writing – original draft:** Robert J. Scheffler, Benjamin P. Bratton, Zemer Gitai.

**Writing – review & editing:** Robert J. Scheffler, Benjamin P. Bratton, Zemer Gitai.

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
