## [Decision Letter · Decision Letter 0]

9 Mar 2022

PONE-D-22-01014Pseudomonas aeruginosa clinical blood isolates display significant phenotypic variabilityPLOS ONE

Dear Dr. Gitai,

Thank you for submitting your manuscript to PLOS ONE. After careful consideration, we feel that it has merit but does not fully meet PLOS ONE’s publication criteria as it currently stands. Therefore, we invite you to submit a revised version of the manuscript that addresses the points raised during the review process.

We look forward to receiving your revised manuscript.

Kind regards,

Chih-Horng Kuo, Ph.D.

Academic Editor

PLOS ONE

Journal Requirements:

Reviewers' comments:

Reviewer's Responses to Questions

**Comments to the Author**

1. Is the manuscript technically sound, and do the data support the conclusions?

Reviewer #1: Partly

Reviewer #2: Partly

2. Has the statistical analysis been performed appropriately and rigorously? 

Reviewer #1: No

Reviewer #2: Yes

3. Have the authors made all data underlying the findings in their manuscript fully available?

Reviewer #1: Yes

Reviewer #2: No

4. Is the manuscript presented in an intelligible fashion and written in standard English?

Reviewer #1: No

Reviewer #2: Yes

5. Review Comments to the Author

Reviewer #1: The manuscript titled “Pseudomonas aeruginosa clinical blood isolates display significant phenotypic variability” has been reviewed. While the study is important, there are substantial concern in the present form of the manuscript. Thus, the present manuscript should be revised.

The manuscript background should be amended, the rationale of the present study, what gap evident in the literature, how the present study mitigates those gaps, what are the specific objectives are to be clearly mentioned.

The authors mentioned that “We obtained 22 clinical blood isolates of P. aeruginosa from different patients from the Robert Wood Johnson Hospital to investigate variability amongst isolates from the same infection site” However, there are no specific details of the samples, The sample description should have proper details. Also, in the methodology, it is mentioned that “All strains were grown at 37 °C in liquid LB Miller (Difco) on a roller drum at 90 rpm.” Are you mentioning the isolates from the samples? No clear details.

What are the controls for each assay? These are not clear. Also, the statistical analysis should mention in the results and discussion. No clear idea about this.

The discussion should emphasize the novelty and uniqueness of the study. What gap mitigated in the literature also need mention, in comparison with latest papers.

The authors mentioned that "There was no significant correlation between the strength of the three phenotypes across isolates, suggesting that they can be independently modulated"". But in the results and discussion it is not easily comprehendible. The author should look into it and revise the manuscript accordingly.

Also, authors mentioned that “Our findings thus demonstrate that no one lab assayed phenotype of pyocyanin production, biofilm production, and twitching motility is necessary for a P. aeruginosa strain to cause blood stream infection and that additional factors may be needed to fully predict what strains will lead to specific human diseases” This is a major point according to this study. According to you, what are the probable reasons and causes, how the same enhances virulence of the organism? There should be separate discussion in these aspects in eth discussion section.

Reviewer #2: This is an interesting paper that describes an important phenomenon that has been a focus of study within the Pseudomonas aeruginosa research community for a number of years now. Genotypic and phenotypic heterogeneity has been described for several clinically relevant niches, none more so than the lungs of patients with Cystic Fibrosis. The current study reports on the phenotypic and genotypic heterogeneity of isolates from blood samples taken from patients and it focuses on three key virulence phenotypes, namely biofilm, twitching and pyocyanin production. The paper is focused and well written.

However, I have some questions relating to the experimental design and interpretation that should be dealt with by the authors. These relate to methodological queries, data availability and the scope of the introduction.

Comments:

The introduction should discuss more about the current state of the art with respect to phenotypic and genotypic heterogeneity with respect to microbial infections, and P. aeruginosa in particular. There are many important papers not cited and they would frame the current paper in a greater context were they to be so.

Page 4, Line 74: Why would the authors expect a ‘stereotypical’ phenotypic response when all the evidence relating to infection in the lungs would suggest that phenotypic and genotypic heterogeneity is common? The interesting finding from this study is that the same applies to bloodstream isolates.

Page 5, Line 95: Interesting that the bloodstream isolates were mucoid biofilms. Can the authors comment on the relevance of this?

Page 6, Lines 113-117: Can the authors be sure that PYO levels in the single isolates reflect the community in the bloodstream? If as they suggest there is significant heterogeneity, then multiple isolates from the same patients and same patient samples would be needed to determine whether or not PYO production was relevant to successful infection. I am not sure the data presented here is sufficient to make this assumption based on a single isolate, albeit from multiple patients. I also have questions relating to the measurement of PYO which I outline below. I am not familiar with this measurement methodology.

Page 7, Lines 154-157: Why would the authors expect the three phenotypes to co-vary? Are they regulated through common systems, is there previous data in the literature to suggest co-variation is likely?

Page 9, Lines193-204: The clustering described in the text is not reflective of what I see in the corresponding figure. Several strains described as clustering with PA14 do not (e.g. RWJ12 and RWJ14), it appears the text describes a different phylogenetic tree?

Page 12, Lines251-253: Can the authors clarify the issues with quality of the genomic assembly scaffolds? If there are issues, how do these impact on the robustness of the data presented?

Page 13, Line 284: Analysis of…………

Figure 1: Have the authors performed statistical analysis on these data and can they show which changes are statistically relevant?

Genome sequencing: Have the authors submitted their genomic data to the e.g. SRA database or other accessible system?

Genome comparison data: Gray boxes indicate that the gene was not found in the scaffolds of the strain and could indicate a loss of the gene or be the result of incomplete assembly of contigs. On this basis can the authors really infer any constructive insight from the analysis, given that the majority of genes were either present or at the end of a contig? Where differences exist, this could be a limitation to the data available? Also, given the comments relating to how regulation could be a factor rather than gene presence/absence, have the authors looked at the promoter regions upstream of the genes studied?

Figure S1: Perhaps I misinterpret the data here but why would PAO1 and PA14 have gray boxes for the phzA-G genes?

Pyocyanin assay: Can the authors provide a reference for the methods used for PYO quantification? The standard assay for pyocyanin analysis would be organic extraction in chloroform followed by 0.2 N HCl. Here the authors have used direct spectrophotometric analysis of the supernatant, which would contain more than one phenazine compound? Is there a reason the organic extraction was not performed, and can the authors clarify that their methodology detects PYO specifically?

6. PLOS authors have the option to publish the peer review history of their article (what does this mean?). If published, this will include your full peer review and any attached files.

Reviewer #1: **Yes: **Sinosh Skariyachan, PhD

Reviewer #2: No

---

## [Author Response · Author response to Decision Letter 0]

27 May 2022

Reviewer #1: The manuscript titled “Pseudomonas aeruginosa clinical blood isolates display significant phenotypic variability” has been reviewed. While the study is important, there are substantial concern in the present form of the manuscript. Thus, the present manuscript should be revised.

Point: The manuscript background should be amended, the rationale of the present study, what gap evident in the literature, how the present study mitigates those gaps, what are the specific objectives are to be clearly mentioned.

Response: We thank the reviewer for pushing us to better clarify the rationale and background and have edited the introduction accordingly.

Point: The authors mentioned that “We obtained 22 clinical blood isolates of P. aeruginosa from different patients from the Robert Wood Johnson Hospital to investigate variability amongst isolates from the same infection site” However, there are no specific details of the samples, The sample description should have proper details. Also, in the methodology, it is mentioned that “All strains were grown at 37 °C in liquid LB Miller (Difco) on a roller drum at 90 rpm.” Are you mentioning the isolates from the samples? No clear details.

Response: We did not isolate the samples ourselves but have attempted to include as many details as possible about the samples. We have also been careful to use the term “strain” instead of “isolate” to make it clear that we did not isolate the strains ourselves in this study.

Point: What are the controls for each assay? These are not clear. 

Response: We have used the well-characterized P. aeruginosa strains PA14 and PAO1, which are each known to be positive for twitching, biofilm formation, and pyocyanin production as positive controls and media as negative controls for our assays. 

Point: Also, the statistical analysis should mention in the results and discussion. No clear idea about this.

Response: As suggested, we have added statistics for some of the other figures and a detailed statistical analysis section to the methods section.

Point: The discussion should emphasize the novelty and uniqueness of the study. What gap mitigated in the literature also need mention, in comparison with latest papers.

Response: We have expanded our discussion to emphasize these important points and contextualize them with other studies.

Point: The authors mentioned that "There was no significant correlation between the strength of the three phenotypes across isolates, suggesting that they can be independently modulated"". But in the results and discussion it is not easily comprehendible. The author should look into it and revise the manuscript accordingly.

Response: We have clarified this point in the text.

Point: Also, authors mentioned that “Our findings thus demonstrate that no one lab assayed phenotype of pyocyanin production, biofilm production, and twitching motility is necessary for a P. aeruginosa strain to cause blood stream infection and that additional factors may be needed to fully predict what strains will lead to specific human diseases” This is a major point according to this study. According to you, what are the probable reasons and causes, how the same enhances virulence of the organism? There should be separate discussion in these aspects in eth discussion section.

Response: This is an important point also touched upon by Reviewer #2. We have thus included in our discussion section a lengthy discussion of the causes, which may relate to different requirements for virulence factors in the blood, and more specifically highlighted the importance for future work beyond the scope of the current study to examine virulence in vivo.

Reviewer #2: This is an interesting paper that describes an important phenomenon that has been a focus of study within the Pseudomonas aeruginosa research community for a number of years now. Genotypic and phenotypic heterogeneity has been described for several clinically relevant niches, none more so than the lungs of patients with Cystic Fibrosis. The current study reports on the phenotypic and genotypic heterogeneity of isolates from blood samples taken from patients and it focuses on three key virulence phenotypes, namely biofilm, twitching and pyocyanin production. The paper is focused and well written.

However, I have some questions relating to the experimental design and interpretation that should be dealt with by the authors. These relate to methodological queries, data availability and the scope of the introduction.

Comments:

Point: The introduction should discuss more about the current state of the art with respect to phenotypic and genotypic heterogeneity with respect to microbial infections, and P. aeruginosa in particular. There are many important papers not cited and they would frame the current paper in a greater context were they to be so.

Response: As mentioned in response to Reviewer #1 we have now expanded the introduction and reframed it to emphasize that others have established virulence factor heterogeneity for P. aeruginosa in other infection sites but that isolates from blood have not been similarly examined. 

Point: Page 4, Line 74: Why would the authors expect a ‘stereotypical’ phenotypic response when all the evidence relating to infection in the lungs would suggest that phenotypic and genotypic heterogeneity is common? The interesting finding from this study is that the same applies to bloodstream isolates.

Response: The reviewer raises a fair point and we have thus edited the text to remove this “straw man” and emphasize the novelty of our findings with respect to blood isolates (whose phenotypic heterogeneity had not been previously examined).

Point: Page 5, Line 95: Interesting that the bloodstream isolates were mucoid biofilms. Can the authors comment on the relevance of this?

Response: This is an interesting question but unfortunately we do not understand its relevance at this point. We thus note that our findings suggest that blood isolates retain mucoid characteristics and that further investigation will be needed to understand its relevance.

Point: Page 6, Lines 113-117: Can the authors be sure that PYO levels in the single isolates reflect the community in the bloodstream? If as they suggest there is significant heterogeneity, then multiple isolates from the same patients and same patient samples would be needed to determine whether or not PYO production was relevant to successful infection. I am not sure the data presented here is sufficient to make this assumption based on a single isolate, albeit from multiple patients. I also have questions relating to the measurement of PYO which I outline below. I am not familiar with this measurement methodology.

Response: We agree that we are measuring PYO levels in a different environment than the blood and from single isolates. We now clarify the caveat that PYO regulation may be different in the bloodstream, but also note that our assay focuses on the capacity of the strains to make this virulence factor in a context in which P. aeruginosa is known to make PYO. We hope that this clarification helps the readers to appreciate what we did, the importance of our findings, and recognize their limitations and importance of future experiments. 

Point: Page 7, Lines 154-157: Why would the authors expect the three phenotypes to co-vary? Are they regulated through common systems, is there previous data in the literature to suggest co-variation is likely?

Response: Several virulence factors are indeed thought to be co-regulated. For example, both biofilm formation and PYO production are regulated by quorum sensing and both biofilm formation and twitching motility are regulated by surface sensing. We have better explained this background in the revision.

Point: Page 9, Lines193-204: The clustering described in the text is not reflective of what I see in the corresponding figure. Several strains described as clustering with PA14 do not (e.g. RWJ12 and RWJ14), it appears the text describes a different phylogenetic tree?

Response: There was an accidental line from a previous version that has been removed and has been double checked against the data presented. We thank the reviewer for catching this error. 

Point: Page 12, Lines251-253: Can the authors clarify the issues with quality of the genomic assembly scaffolds? If there are issues, how do these impact on the robustness of the data presented?

Response: Like most genomic assemblies, our assemblies are incomplete due to the length of our reads and the depth of coverage. This does not affect the data presented as we accounted for representation issues in our analysis, as explained in the methods section and the caption to Fig. 4. 

Point: Page 13, Line 284: Analysis of…………

Response: We forget the word “of” and fixed that issue.

Point: Figure 1: Have the authors performed statistical analysis on these data and can they show which changes are statistically relevant?

Response: As suggested, we have statistically compared all of our isolates to our PAO1 control and now show that some are statistically similar and some significantly distinguishable from PAO1.

Point: Genome sequencing: Have the authors submitted their genomic data to the e.g. SRA database or other accessible system?

Response: The data have been submitted to the NIH Genbank, we included the accession in the Assembly Statistics Table and have included it in the revised text. 

Point: Genome comparison data: Gray boxes indicate that the gene was not found in the scaffolds of the strain and could indicate a loss of the gene or be the result of incomplete assembly of contigs. On this basis can the authors really infer any constructive insight from the analysis, given that the majority of genes were either present or at the end of a contig? Where differences exist, this could be a limitation to the data available? Also, given the comments relating to how regulation could be a factor rather than gene presence/absence, have the authors looked at the promoter regions upstream of the genes studied?

Response: First, it is important to clarify that the fully Gray boxes are the only genes that were not found. Checkerboard patterns reflect genes that were found but were at the end of contigs and thus partial sequences, but we still were able to determine the extent of gene similarity to these partial reads. Thus, for the majority of genes we can indeed provide insight. We agree with the reviewer that one should be cautious to draw strong conclusions about a lack of finding a particular protein and why we focus on the high level of protein similarity we find between the strains.

We have not looked at promoters as our analysis of protein similarity within the assay associated pathways was performed at the level of protein sequence instead of DNA, and not all of these genes have clearly defined promoters. Additionally, given that some proteins were at the ends of the contigs, these would be entirely lacking promoters, even when a portion of the coding sequence was identified.

Point: Figure S1: Perhaps I misinterpret the data here but why would PAO1 and PA14 have gray boxes for the phzA-G genes?

Response: To provide a more fair comparison we included PAO1 and PA14 in our genomic sequencing and assembly pipeline and for our analysis compared our de novo assembly to the published annotated genomes. This gives us a baseline for understanding the frequency of missing genes. We have clarified this point in the revised text.

Point: Pyocyanin assay: Can the authors provide a reference for the methods used for PYO quantification? The standard assay for pyocyanin analysis would be organic extraction in chloroform followed by 0.2 N HCl. Here the authors have used direct spectrophotometric analysis of the supernatant, which would contain more than one phenazine compound? Is there a reason the organic extraction was not performed, and can the authors clarify that their methodology detects PYO specifically?

Response: In P. aeruginosa, PhzS is responsible for making pyocyanin and 1-hydroxyphenazine. By comparing the UV-VIS spectra from WT and Tn::phzS, strains we saw a reduction in the absorbance around 320 nm indicating that the peak measured is pyocyanin as phzS is responsible for making pyocyanin. Thus, we used the absorbance in this region as a correlate for determining pyocyanin levels without performing a full extraction on all strains tested.

---

## [Decision Letter · Decision Letter 1]

14 Jun 2022

Pseudomonas aeruginosa clinical blood isolates display significant phenotypic variability

PONE-D-22-01014R1

Dear Dr. Gitai,

We’re pleased to inform you that your manuscript has been judged scientifically suitable for publication and will be formally accepted for publication once it meets all outstanding technical requirements.

Kind regards,

Chih-Horng Kuo, Ph.D.

Academic Editor

PLOS ONE

Additional Editor Comments (optional):

Congratulations on the successful revision. The reviewers have some more minor suggestions, which I consider to be optional. Also, there are some minor mistakes, such as extra line breaks in lines 204/229/391. Please make all necessary changes and send the final version to editorial office for production, as well as arrange for data release in GenBank.

Reviewers' comments:

Reviewer's Responses to Questions

**Comments to the Author**

1. If the authors have adequately addressed your comments raised in a previous round of review and you feel that this manuscript is now acceptable for publication, you may indicate that here to bypass the “Comments to the Author” section, enter your conflict of interest statement in the “Confidential to Editor” section, and submit your "Accept" recommendation.

Reviewer #1: (No Response)

Reviewer #2: All comments have been addressed

2. Is the manuscript technically sound, and do the data support the conclusions?

Reviewer #1: Yes

Reviewer #2: Yes

3. Has the statistical analysis been performed appropriately and rigorously? 

Reviewer #1: Yes

Reviewer #2: Yes

4. Have the authors made all data underlying the findings in their manuscript fully available?

Reviewer #1: Yes

Reviewer #2: Yes

5. Is the manuscript presented in an intelligible fashion and written in standard English?

Reviewer #1: Yes

Reviewer #2: Yes

6. Review Comments to the Author

Reviewer #1: (No Response)

Reviewer #2: The authors have carefully considered all points made in the initial peer review, have provided a rational and coherent response to all points made, and have revised the manuscript accordingly.

I just have two points to complete the review: Firstly, the comment on mucoidy of blood isolates refers to the fact that mucoid Pa strains are often an indication of a chronic persistent infection. One would imagine that the blood stream isolates would typically be more associated with an acute infection and therefore would not typically possess the mucoid phenotype. An interesting observation that does not need to be addressed, just a point of note. Secondly, the authors provide some good evidence for the PYO measurements, and the increased PYO levels in PA14 compared to PAO1 appear to support their methodology. It would be good for the authors to consider a methods paper that validates their detection method using (i) PYO (commercially available) and/or (ii) extracts from Pa isolates. Their methodology would enable a higher throughput analysis of PYO production in Pa, but further validation would be needed to justify its adoption as a core method.

7. PLOS authors have the option to publish the peer review history of their article (what does this mean?). If published, this will include your full peer review and any attached files.

Reviewer #1: No

Reviewer #2: No

---

## [Editor Report · Acceptance letter]

27 Jun 2022

PONE-D-22-01014R1 

*Pseudomonas aeruginosa* clinical blood isolates display significant phenotypic variability 

Dear Dr. Gitai:

I'm pleased to inform you that your manuscript has been deemed suitable for publication in PLOS ONE. Congratulations! Your manuscript is now with our production department. 

Kind regards, 

on behalf of

Dr. Chih-Horng Kuo 

Academic Editor

PLOS ONE